# Silicone vs. Polyurethane Stent: The Final Countdown

**DOI:** 10.3390/jcm11102746

**Published:** 2022-05-12

**Authors:** Suresh Mathias, Oliver Wiseman

**Affiliations:** 1Locum Consultant Urologist, Royal Wolverhampton NHS Trust, Wolverhampton WV10 0QP, UK; s.mathias1@nhs.net; 2Consultant Urologist, Cambridge University Teaching Hospitals NHS Trust, Cambridge CB2 0QQ, UK

**Keywords:** ureteral stents, stent-associated symptoms, encrustation, biofilm

## Abstract

Ureteric stents are conventionally used in daily urological practice. There is ongoing debate on the superiority of different stent materials, particularly in terms of patient tolerance. We conducted a literature review to compare silicone stents and stents made of other materials from a patient tolerability perspective. We conclude that silicone stents are better tolerated but further research is required.

## 1. Introduction

Since being introduced by Finney in 1978 [1], ureteric stents are frequently used in endourology, to assist in the temporary or permanent drainage of the kidney. They allow the urine to flow down past a blockage impairing its drainage, which may be either internal or external, and thus help to ensure optimum renal function. They are used frequently in the peri-operative management of urolithiasis, often being placed post ureteroscopic procedures [2], although RCTs have shown that routine stenting perhaps is not needed after an uncomplicated ureteroscopy (URS) [3,4,5,6]. EAU guidelines state that stents need not be used in such cases [7]. This guideline states that stents should be inserted in patients who are at increased risk of complications, such as those with ureteral trauma, residual fragments, bleeding, perforation, UTIs, or in those who are pregnant, and in all doubtful cases, to avoid stressful emergencies. The ideal duration of stenting post ureteroscopy is not known, with most urologists favouring up to two weeks depending on the exact indication for placement. Stents may also be placed to allow healing of a ureteric injury, or after a vesico-ureteric anastomosis, such as would be needed after a renal transplant. Alpha-blockers have been shown to reduce the morbidity of ureteral stents [8,9].

Placement of a stent prior to ureteroscopic management of a stone may facilitate ureteroscopic management of such stones, improve the stone free rates (SFR), and reduce intra-operative complications [10,11]. However, the driver for limiting the usage of stents is due to the significant negative effect that they can have on a patient’s quality of life. These effects include pain on urination, troublesome urinary symptoms, haematuria, as well as increased infection risk, biofilm formation and stent encrustation [12]. The formation of a biofilm may be an important aetiological factor in infection and sepsis after endourological manipulation in a patient with an indwelling stent, and encrustation of a stent is a significant complication which often requires multi-modality surgery to resolve.

Stent-associated symptoms are thought to be due to a combination of localised inflammation, elevated bladder pressure causing reflux of urine towards the kidney, reduced ureteric peristalsis and bladder irritation. The three factors which affect stent performance are their design, the material they are made of, and any surface coating they may have.

## 2. Stent Design

The most common stent design, introduced by Finney, is a “double-J”. It is so called because each end of the stent forms a J, effectively keeping the stent in place, with one “J” sitting in the bladder, and the other one in the renal pelvis of the kidney. Several design changes have been introduced, with the aim of trying to reduce symptoms for patients as well as biofilm formation and encrustation on the stent. Stents with a spiral cut have been studied and introduced, with the aim of improving the flow of urine and conforming to the shape of the ureter, thus leading to a decrease in patient symptoms, although an in vivo study on pigs showed no difference in encrustation, infection or stent migration [13]. Stents have been designed with side holes, to try and improve drainage of urine, with stents which have a greater number of side holes having better drainage [14]. Other design innovations have included the introduction of a tail stent, which is similar to a double-J stent, except that the lower end of the stent has no “J”, and may be made of a different softer material, to the rest of the stent, thus decreasing bladder irritation. One clinical study of such a stent showed less bladder irritation compared to the traditional stent [15]. However, another study comparing two stents, one a pigtail stent, the Polaris^TM^ stent (Boston Scientific, Marlborough, MA, USA), which is made of a Percuflex^®^ combination, providing a firm proximal aspect with a softer distal aspect to minimize symptoms, and one a standard “double-J”, the Inlay^®^ stent (Bard Medical, Murray Hill, NJ, USA) in 98 patients showed no difference in urinary symptom score between the two groups [16]. Another design innovation is of dual-durometer stents where the material of the stent transitions, and differs, from being harder at the proximal end to being softer towards the distal (bladder) end. This type of stent design has not been formally studied.

## 3. Stent Material

As well as stent design, another important aspect which has been studied has been stent material. The material which is used needs to have a number of key properties, including biocompatibility, mechanical strength, flexibility, radio-opacity, lack of degradation in the urinary tract, a long shelf-life, and cost-effectiveness.

Some of the earliest stents were made of polyethylene, a synthetic polymer [17] and had a single loop which led to positioning difficulties, and proximal migration. The double pigtail version performed better. Polyurethane was the next material to be used but tended to be associated with stent fracture. This was observed to occur along the drainage holes of the stent, suggesting that elimination of these holes would reduce the incidence of polyurethane ureteral stent fracture in use.

Another early material to be investigated was silicone. Silicone is extremely biocompatible [18] and, due to its softness, was thought to be associated with less patient discomfort [19,20]. The hydrophilic coating offers a better glide for stent insertion, and additionally it was thought to decrease encrustation [21].

Despite the early findings in relation to silicone as a material for JJ stents, until recently, no high-level evidence has existed to affirm or refute these findings, and indeed no study had shown any convincing evidence of the benefit of one type of stent over another in relation to patient comfort or tendency to encrustation.

Softness of stent material has been examined by other investigators. The Contour ureteral stent, a soft polymer, was compared in a randomised control study to the Percuflex stent, in 130 patients needing stent placement during the treatment of renal stones. The USSQ (Ureteral Stent Symptom Questionnaire) was completed at 1 and 4 weeks with the stent in place, and there were no significant differences seen in the domains of pain, urinary symptoms, or general health. Additionally, there were no differences seen between the groups in the number of days with reduced activities, or in work performance or sexual dysfunction [22].

One factor which is not inherently linked to design is that of choosing the correct length of stent. One study by Taguchi showed that the length of a ureteric stent is an important factor to reduce stent symptoms [19], with stents which crossed the midline contributing to worse stent-associated symptoms.

## 4. Evidence Regarding Tolerability of Silicone Stents

The superiority of silicone ureteric stents in minimising stent-associated symptoms after flexible ureteroscopy for renal stone disease has been shown in a large multicentre, single-blinded, prospective RCT by Wiseman et al. [23]. The primary outcome was elucidated with the Ureteral Stent Symptom Questionnaire (USSQ) Body Pain Index at day 20, prior to stent removal, and the secondary endpoints were USSQ scores addressed on days 2, 7 and 20 post-operatively, and at 2 weeks post stent removal. Adverse effects were also documented.

This study was performed in 4 different hospitals, across three countries. Patients who had renal stones between 5 and 25 mm were included, and the exclusion criteria were those patients with anatomical malformations of renal tract, urogenital tumours, patients with pre-stented ureters and untreated UTIs. Comparison was made between the hydrocoated silicone stent (Coloplast Imajin^TM^ hydro, Coloplast, Humlebaek, Denmark) and the Percuflex Plus^TM^ stent (Boston Scientific, Marlborough, MA, USA) with regards to patient comfort and quality of life (QOL) after flexible ureteroscopy (FURS) for renal stone disease over a 5-week prospective follow-up. The allocated stent was checked from the random list just before stent insertion, with a 1:1 allocation.

Of note, 141 patients who underwent FURS for renal stone disease were included in this study, with 113 (80.1%) patients completing the study. Patients enrolled in the study were identical with respect to baseline characteristics of age, gender, BMI, the number having their first stone episode, the number with a symptomatic stone, and procedure duration. Predominant withdrawals from this study were due to patient noncompliance, lost to follow-up or patients who did not want to participate further in the trial. Sixty-eight patients were randomised to the Imajin^TM^ hydro silicone stent and 73 to the Percuflex Plus^TM^ stent. This is shown in the Consort Diagram, shown in Figure 1. The timing for stent removal by flexible cystoscopy was scheduled for 20 days and the patients were followed for 2 further weeks.

Mean USSQ body pain index score at day 20 observed for the silicone stent was 18.7 (11.40) vs. 25.1 (14.2) for the Percuflex Plus^TM^ stent (*p* = 0.015). USSQ urinary symptoms score observed in the silicone stent group at D20 were 26.4 (7.7) vs. 31.8 (8.1) (*p* < 0.001) in the Percuflex Plus^TM^ stent group. This represented a significant reduction in the USSQ pain score, and this improvement was maintained even when the USSQ scores were normalised for the differences in gender frequencies between the two groups, as the Silicone group had more women compared to the Percuflex plus group. While the primary endpoint of the study showed this benefit for silicone, additional benefit was seen when looking at the secondary outcomes, with significantly lower scores seen in the Urinary symptom domain of the USSQ at day 2, 7 and 20, with the greatest difference being seen at day 20, with the mean (SD) score of 26.4 (7.7) vs. 31.8 (8.1) in favour of the silicone group (*p* < 0.001).

A further study has backed up these findings. Gadzhiev et al. enrolled 50 patients from two centres in a non-randomised fashion, who underwent stent placement after presenting with ureteric colic. They received either a polyurethane stent (Rusch, Teleflex), or a silicone stent (Cook Medical). They showed that, compared to polyurethane ureteral stents, silicone ureteral stents were associated with lower body pain intensity assessed by visual analogue scale for pain 2 weeks before stent removal, and at the time of stent removal [24].

The evidence therefore clearly demonstrates that silicone stents are better tolerated than other stent types, and specifically better than the Percuflex Plus^TM^ stent and polyurethane stent, respectively.

## 5. Evidence Regarding Encrustation of Silicone Stents

The same prospective randomized multicenter, single-blinded clinical comparative study as above, and published by Barghouthy et al. [25], showed that silicone-hydrocoated stents (Coloplast Imajin^TM^ hydro) are less prone to encrustation than the Percuflex^TM^ Plus stent (Boston Scientific) after a 3-week indwell period, and this confirms the low encrustation potential of silicone.

The study included 141 patients as outlined previously. Endpoints related to encrustation were biofilm formation and mineral encrustation after a 3-week indwell time. They were evaluated at removal through a scoring scale of ureteral stent encrustation, using infrared spectroscopy and optical microscopy of inner and outer surfaces of tips, angles and along the stent body. This is shown in Table 1. Comparison was performed using ANOVA.

In total, 119 stents were analysed after removal, 56 in the silicone and 63 in the Percuflex^TM^ Plus Group. Mean indwelling time was 21.8 days for the silicone group and 22.1 days for the Percuflex^TM^ Plus group. There was significantly more biofilm on the Percuflex^TM^ Plus stents compared to the silicone stents (1.24 ± 0.08 vs. 0.93 ± 0.09, *p* = 0.0021), as well as more mineral encrustation (1.22 ± 0.10 vs. 0.78 ± 0.11, *p* = 0.0048), respectively. The study concluded that silicone stents are therefore less prone to encrustation and biofilm formation than Percuflex Plus^TM^ stents.

Another study which corroborates these findings was undertaken by Tunney [27]. This study compared stents made of five different polymeric materials for encrustation, after a 14-week period of suspension in an artificial urine bath. The study showed that silicone had the best performance in the long term, showing 30% less encrustation at 10 weeks.

## 6. Recent Evolutions of Stent Design

Another recent RCT [28] suggests that using a “pigtail suture stent” (PSS) called the J-Fil (Rocamed, Monaco), with a fluted beak at the distal end attached to a suture rather than a “J” curve, of a conventional stent may reduce stent-associated symptoms.

This prospective, single-blind RCT compared stent-related symptoms caused by the PSS with a conventional double-J stent (Vortek, Coloplast, Humlebaek, Denmark) after uncomplicated ureteroscopy for stone disease. In this study, patients undergoing semirigid or flexible ureteroscopy for ureteral or renal stones <2 cm were randomised. Patients with distal ureteric stones were excluded, as were those who had a JJ stent prior to the procedure.

Two weeks after surgery, patients having the PSS had better outcomes compared to the control group in the measures of Urinary Symptom Index score (from a validated Italian translation of the USSQ), 24 vs. 30 (*p* = 0.004), overall VAS score, 2 vs. 4 (*p* = 0.02), and the percentage of patients complaining of body pain and discomfort (64% vs. 86%; *p* = 0.03). While this innovation in stent design would appear to show a significant benefit when placed post ureteroscopy, there are two issues to highlight. The first is that, comparison to the Vortek stent, which is not known for being a soft material, does not mean that the PSS will be better tolerated than, for example, a silicone stent. The second is that patients with distal ureteric stones were excluded in this study, and this patient group represents a significant proportion of patients who would undergo a ureteroscopy, meaning that the findings may not be applicable to all post-ureteroscopy patients.

A further recent innovation in stent design is the Tria^TM^ Soft Ureteral Stent (Boston Scientific, Marlborough, MA, USA). This encompasses PercuShield^TM^ technology on both the outer and inner surfaces of the stent, in order to try and reduce the accumulation of magnesium and calcium deposits. While this stent is in clinical use, short-term data comparing it to the Polaris Ultra stent showed no difference in preventing encrustation. Long-term data are needed to determine whether this new material confers any benefit in clinical practice [29].

## 7. Discussion

Ureteric stents are frequently used in the management of urolithiasis and are associated with a considerable impact on a patient’s quality of life. Improvements in stent design and the use of different materials have been evolving to try and reduce these symptoms, as well as the risk of biofilm formation and stent encrustation. Based on the studies discussed in this paper, we would suggest that the silicone stent is an excellent option for the post-operative drainage of patients who have undergone a flexible ureteroscopy for stone disease. They are associated with less post-operative pain and fewer urinary symptoms at day 20 post-op. The reasons for these differences are likely related to the material itself. Silicone is biocompatible and soft. However, it is clear from the previously mentioned study by Joshi et al. that softness of the material may not be the sole determinant of stent comfort. Other as-yet-unknown factors may be critical. Regarding encrustation potential, the randomised evidence would support the view that silicone, as a material for stent manufacture, has a lower encrustation potential.

However, it should be noted that, due to the softness of silicone as a material, it may not perform to provide adequate drainage in situations of ureteric stricture or external compression. The studies which have been discussed above were undertaken in patients with urolithiasis. There are a number of options in such situations, ranging from specific tumour stents, such as that from Coloplast, which has reinforced internal layer for excellent resistance to compression, to metallic stents, such as the Resonance^TM^, Allium^TM^ or Memokath^TM^ ureteral stents, to extra-anatomic stents such as the Detour^TM^ bypass stent. Ureteral obstruction can be benign (BUO) or malignant (MUO), and in a systematic review of long duration stents for ureteral stricture undertaken by Corrales et al. [30], the Resonance^TM^, Allium^TM^ and Memokath^TM^ ureteral stents were found to be useful for both BUO and MUO. The Resonance stent (Cook, Bloomington, IN, USA) is made of a cobalt-chromium-nickel-molybdenum alloy (MP35N). An internal wire (also made from MP35N) extends the full length of the stent and joins the stent at either extremity and this wire prevents the elastic elongation of the stent, which is particularly important during stent removal. The tightly wound coil design helps maintain continuous drainage by allowing urine to flow in and out of the coils. The Resonance^TM^ stent has an indwell time of up to one year. The Allium^TM^ stent (Allium, Caesarea, Israel) is a self-expanding large caliber stent, and comes in sizes from 24 Fr to 30 Fr in diameter. It is made of a super elastic nitinol alloy covered by a unique polymer and has an indwell time of up to three years. The MemoKath^TM^ stent (PNN, Kvistgård, Denmark) is made of a thermo-expandable nickel-titanium alloy with memory-shape effect and has a tight spiral structure which allows it to conform and adapt to the natural curves of the ureter, while making tissue ingrowth between the coils difficult. Memokath^TM^ is inserted unexpanded into the desired location in the ureter, and, when expanded, the ends anchor the stent in place. All of these metallic stents can be placed antegradely or retrogradely, and although relatively highly priced compared to standard stents, they can show a financial advantage over the long term.

The Uventa^TM^ stent (TaeWoong Medical, Seoul, Korea) is a self-expanding covered metallic stent. It has a three-layered construction with the outer layer having a high friction coefficient to prevent migration, and the inner layer reinforcing the overall radial force. A PTFE membrane prevents tissue ingrowth. This stent was found to be a good option in patients with chronic ureteral obstruction. The Detour bypass stent^TM^ (Coloplast, Humlebaek, Denmark) is a possible option for patients with complete obstruction of the ureter, who are unfit for reconstructive surgery. A Detour stent consists of an outer layer made of polytetrafluoroethylene and an inner layer that is a silicone tube 17 F wide, with perforations on both ends. The Detour is placed subcutaneously to the kidney and to the bladder. The route is then tunnelled by the large plastic hollow tube for inserting the Detour, and its distal end is sutured to the bladder. Finally, although tumour stents may provide good drainage, there are very few published studies on it.

## 8. Conclusions

The ideal ureteric stent should possess the following qualities: be easy to insert and remove; have good memory to minimise migration; have excellent flow characteristics; be radio-opaque; be biologically inert; resist biofilm formation; encrustation and thereby stent-associated infection; be flexible with good tensile strength; be reasonably priced; and, last but by no means least, cause minimal symptoms and complications.

With respect to these qualities, silicone stents have now been shown to be better tolerated and to be less prone to biofilm formation and encrustation. There are further innovations in stent design, which will need further evaluation, but it is clear that, at the present time, if stent placement post ureteroscopy cannot be avoided, a silicone stent is an excellent choice. Thus while “The Final Countdown” may not yet have reached zero, as we approach it, silicone is in the lead as the best tolerated and least likely to encrust material from which to make a stent.

## Figures and Tables

**Figure 1 jcm-11-02746-f001:**
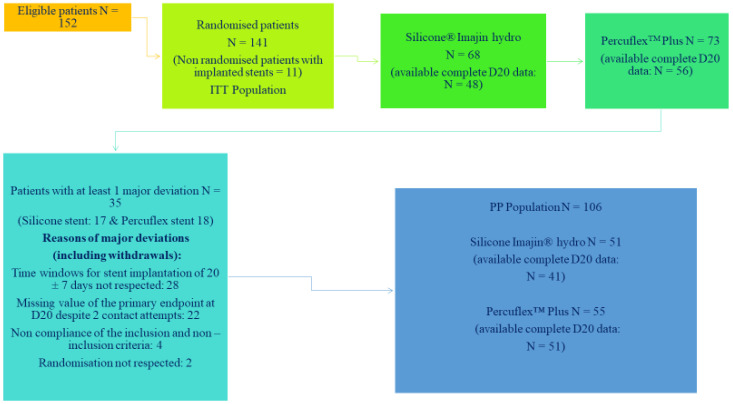
CONSORT Diagram of analysed population [23].

**Table 1 jcm-11-02746-t001:** Scale for scoring of encrustations and biofilm on double-J stents [26].

	Scale for Scoring Biofilm	Scale for Scoring Mineral Encrustation
0	No biofilm	No mineral encrustation
1	Small, well-circumscribed plaques of fine organic encrustation, especially at stent orifices	Few crystals
2	More or less extensive plaques of fine organic film	Some crystals
3	Fine organic film covering part of outer and inner surface of stent, and clogging some orifices within this section of stent	Fairly many, locally confluent mineral encrustations
4	Extensive organic film covering part of outer and inner surface of the stent, and nearly completely clogging orifices within this section of stent	Many confluent, but thin mineral encrustations
5	Extensive organic film covering part of outer and inner surface of the stent, clogging several orifices, and even partially or completely occluding lumen	Extensive, thick in places (≥1 mm), surrounding part of stent
6		Stone formed at end or on body of stent

## Data Availability

Not applicable.

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
