# Peer review of "Silicone vs. Polyurethane Stent: The Final Countdown"

_jcm, 2022, doi:10.3390/jcm11102746_

Round 1

Reviewer 1 Report

  • General comments

The authors conducted a literature review to compare silicone stents and stents made of other materials from a patient tolerability perspective. They concluded that silicone stents have now been shown to be better tolerated and to be less prone to biofilm formation and encrustation.

The reviewer generally agrees with the conclusion.

However, there are several issues need to improve. The reviewer would like suggests several issues as follows;

  • Specific comments for revision
  1. Minor

#1 Please describe any differences in stone adhesion depending on the stent material?

#2 Please describe any differences in stent lumen obstruction depending on the stent material?

#3 Please describe any differences in stent function with and without side holes?

Author Response

Thank you for your review. We have addressed your specific comments as follows.

#1 Please describe any differences in stone adhesion depending on the stent material?

This has been done in the paper, and the study reference 25 (Barghouthy Y, Wiseman O, Ventimiglia E., et al. Silicone-hydrocoated ureteral stents encrustation and biofilm formation after 3-week dwell time: results of a prospective randomized multicenter clinical study; World J Urol. 2021 Sep;39(9):3623-3629. doi: 10.1007/s00345-021-03646-0. Epub 2021 Mar 10), which is discussed in section 5, covers this.

#2 Please describe any differences in stent lumen obstruction depending on the stent material?

In section 5 we cover the aspects of the stent which were examined, and again further details regarding this is available from reference 25. The scoring system for assessing these stents is referenced in reference 33, and both inner and outer surfaces of the stent.

#3 Please describe any differences in stent function with and without side holes?

This is not the prime focus of the paper, which is on stent material, and there is very little evidence regarding this. However, we have annotated the text and included the following reference which we hope that readers will find helpful should they wish to explore further. Thank you for bringing this to our attention.

Kyung-Wuk Kim Young Ho Choi  Seung Bae Lee Yasutaka Baba  Hyoung-Ho Kim Sang-Ho Suh. Numerical analysis of the effect of side holes of a double J stent on flow rate and pattern. Biomed Mater Eng 2015;26 Suppl 1:S319-27.

Reviewer 2 Report

Dear Editor and Author, In my opinion, whole paper does not raise any substantive doubts.
It is written in clear and transparent scientific language.
All abbreviations are clearly explained.
The methodology, results and conclusions are detailed. As a clinician urologist I have same observation about tolerance for both kinds of stents. I fully recommend.  Regards

Author Response

Thank you ever so much for your kind review

Reviewer 3 Report

Dear Authors, the paper is well design and synthesized. Self-citation is a good option for the logical construction of the theme. Good excursus about all market solutions about stenting in different situations. It's probably premature to think about the "countdown" of all type of poliurethane stents in term of tolerability and discomfort. e.g. J-Fil Stent  could be a good option in different scenarios with a long indwelling time life. The good tolerability & biocompatibility and less tendency to encrustation of silicon stents is scientifically known but we still lack data on new materials like Tria° stent.

The Poliurethane road of sunset? we need more randomized trials on this focus

Author Response

Many thanks for your kind review. We completely agree that more trials are needed to establish the ideal ureteric stent. We discuss the new designs in our paper, and conclude that further studies are required, so we think that we are in agreement with the reviewer. We have also concluded similarly to the reviewer, in stating that “The Final Countdown” may not have reached zero yet.

Reviewer 4 Report

Well written review,  and good focused on silicone and polyurethrane stents. The authors should also mention about benefits of ureteral stent replacement during renal transplantation. The authors can add the studies below as a references to increase the value of content introduction section.

Yuksel Y, Tekin S, Yuksel D, Duman I, Sarier M, Yucetin L, Kiraz K, Demirbas M, Kaya A.F., Sezer M.A, Demirbas A. and Yavuz A.H. Optimal Timing for Removal of the Double-J Stent After Kidney Transplantation. Transplant Proc. 2017;49(3):523-527. doi:10.1016/j.transproceed.2017.01.008

Sarier M, Seyman D, Tekin S, et al. Comparision of Ureteral Stent Colonization Between Deceased and Live Donor Renal Transplant Recipients. Transplant Proc. 2017;49(9):2082-2085. doi:10.1016/j.transproceed.2017.09.028

Author Response

Thanks very much for your kind review. Bringing into the focus the other situation in which stents might be used is certainly helpful, and we are grateful to the reviewer for bringing this to our attention. We have added a sentence in the introduction to highlight this, which states: “Stents may also be placed to allow healing of a ureteric injury, or after a vesico-ureteric anastomosis, such as would be needed after a renal transplant”. We do not feel that adding a reference to either of the suggested papers will make this aspect of the paper more robust, so have elected not to include a reference at this stage.

Round 2

Reviewer 4 Report

well written study. good focused on ureteral stents.

This manuscript is a resubmission of an earlier submission. The following is a list of the peer review reports and author responses from that submission.